# Evaluation of Different Techniques, including Modified Atmosphere, under Vacuum Packaging, Washing, and *Latilactobacillus sakei* as a Bioprotective Agent, to Increase the Shelf-Life of Fresh Gutted Sea Bass (*Dicentrarchus labrax*) and Sea Bream (*Sparus aurata*) Stored at 6 ± 2 °C

**DOI:** 10.3390/biology11020217

**Published:** 2022-01-29

**Authors:** Lucilla Iacumin, Am Stefania Jayasinghe, Michela Pellegrini, Giuseppe Comi

**Affiliations:** Department of Agricultural, Food, Environmental and Animal Science, University of Udine, Via Sondrio 2/a, 33100 Udine, Italy; lucilla.iacumin@uniud.it (L.I.); jayasinghe.amstefania@spes.uniud.it (A.S.J.); pellegrini.michela@spes.uniud.it (M.P.)

**Keywords:** fish, biopreservation, *Latilactobacillus sakei*, packaging, storage temperature, shelf-life

## Abstract

**Simple Summary:**

A method of increasing the shelf life of gutted sea bass and sea bream packaged under vacuum and stored at 6 ± 2 °C (simulating supermarkets and consumer fridges) through the use of bioprotective starter cultures was studied. An increase in the shelf life up until 14 days was observed by washing the gutted fish in water supplemented with a culture of *Latilactobacillus sakei* and dextrose (0.1%). *L. sakei* reduced the growth of specific spoilage microorganisms and consequently reduced the production of total volatile nitrogen and oxidized compounds.

**Abstract:**

Fish meat is very perishable because of indigenous and microbial enzymes, which determine spoilage and shelf life. The deterioration processes, which lead to an important, sequential, and progressive modification of the initial state of freshness, are fast and depend on rearing, harvesting, slaughtering, handling, and storage conditions. Usually, the shelf life of gutted fish stored at 4 ± 2 °C under vacuum packaging (VP—1.0 bar) and modified atmosphere packaging (MAP, 70% N_2_, <1% O_2_, 30% CO_2_) is approximately 9 days. The aim of this work was to improve the shelf life and preserve the microbiological and sensory quality of farmed gutted sea bass (*Dicentrarchus labrax*) and sea bream (*Sparus aurata*) using different methods, including VP, MAP, and bioprotective culture containing *Latilactobacillus sakei*, until 12–14 days. Microbiological, physicochemical, and sensory quality indices were monitored to confirm the effectiveness of biopreservation on product quality during proper refrigeration (4 ± 2 °C) or abuse (6 ± 2 °C, simulating supermarkets and consumer fridges) storage period. Considering the quality indexes represented by *Enterobacteriaceae*, total volatile nitrogen (TVB-N), and malonaldehyde concentrations (TBARS) and the sensorial analysis, the VP samples were more acceptable than the MAP fish, even though the shelf-life of the VP and MAP fish was similar at about 12 days. The second phase of the work was to evaluate the shelf-life of both VP fish stored at 6 ± 2 °C, which simulates the normal abuse temperature of supermarkets or consumer fridges. Data confirmed the previous results and demonstrated, despite the abuse temperature of storage, a shelf-life of about 12 days. Finally, the third phase consisted of prolonging the shelf life until 14 days of storage at 6 ± 2 °C by washing the gutted sea bass and sea bream in a suspension of bioprotective starter (7 log CFU/mL) with or without the addition of dextrose (0.1%) and by VP packaging. The bioprotective culture reduced the growth of spoilage microorganisms. Consequently, the total volatile nitrogen (TVB-N) concentration in both fish species was low (<35 mg N/100 g). Nonprofessional and untrained evaluators confirmed the acceptability of the inoculated samples by sensorial analysis.

## 1. Introduction

Sea bass (*Dicentrarchus labrax*) and sea bream (*Sparus aurata*) are the main marine fish species farmed in Europe and, in particular, in Mediterranean countries. Their white flesh, low fat and high content of polyunsaturated fatty acids (PUFAs), mainly eicosapentaenoic acid (EPA) and docosahexaenoic acid (DHA), make them popular and the most important economically cultured fish among aqua-cultured species [1,2,3,4,5]. Indeed, the growing interest of consumers in nutritional aspects and the parallel attention to food quality issues have contributed to their consideration as a value-added seafood product, with increasing consumption and demand in the international seafood market [1,3]. For this reason, great interest has been given to rearing systems and feeding regimes because they may affect flesh quality, especially in terms of fat concentration, nutritional compounds, and hygienic quality [6,7]. Both fish are very perishable because of indigenous and microbial enzymes, which determine deterioration and shelf life [8] immediately postmortem [9]. The deterioration processes, which lead to an important, sequential, and progressive modification of the initial state of freshness, are fast and depend on rearing, harvesting, slaughtering, handling, and storage conditions [9,10,11].

Microbial spoilage is attributed to specific spoilage organisms (SSOs) that prevail over the rest of the microbiota, reach high concentrations (>8 log CFU/cm^2^ or g), are favoured by different parameters (such as atmosphere, temperature, and flesh chemical composition; processing, transportation, and storage in the market) and produce various metabolites responsible for off-flavours/odours [8,9,12,13].

Aerobic and facultative anaerobic Gram-negative (*Pseudomonas*, *Moraxella/Acinetobacter*, *Photobacterium*, *Flavobacterium/Cytophaga*, *Xanthomonas*, *Vibrio*, *Shewanella*, *Proteus*, *Aeromonas*, *Serratia*, *Hafnia*) or Gram-positive (*Bacillus*, *Corynebacterium*, *Micrococcus*, *Carnobacterium,* and other cocci and lactobacilli) microorganisms can grow during storage and are the main bacteria isolated in spoiled fish that are stored in air or modified atmosphere packaging (MAP) [14,15]. *Photobacterium phosphoreum* is a spoiler of cold water fish, but it is also present in Mediterranean species [15,16,17,18,19].

The spoilage of fish products includes the production of short-chain peptides, amino acids and other nonprotein nitrogen molecules [20,21,22,23], trimethylamine (TMA), total volatile nitrogen (TVB-N), sulfuric compounds, aldehydes, ketones and esters [2,3,4,8], lactic acid, acetic acid, ethanol, hydrogen sulfide, thiols, mercaptans, dimethyl sulfides, and indole, which produce urinary odours in fish meat [24,25,26,27,28,29,30,31]. When aerobically stored, fresh fish are particularly spoiled by *Pseudomonas* spp., producing TVB-N [25,31], which increases at the end of storage. Conversely, when stored in MAP, TVB-N development is slower [2,3] because *Pseudomonas* growth is suppressed, and lactic acid bacteria (LAB) and *Brochothrix thermosphacta* predominate and produce organic acid and volatile compounds [32]. Finally, *Hafnia alvei*, *Proteus* spp., *Pseudomonas* spp., *Shewanella putrefaciens* and *Morganella morganii* [17,33,34] decarboxylate amino acids to biogenic amines, which represents a risk for consumers [35,36,37]. An increase in the amount of these compounds is suitable only as an acceptance/rejection criterion and is not suitable as a freshness index [24,25].

Different and current technologies, including refrigeration of the products after air or MAP or adding natural preservatives (e.g., essential oils), have been used in order to increase the shelf-life of sea bass and sea bream fish. Unfortunately, these strategies do not permit the entire control of spoilage bacteria. Therefore, new technologies are needed to increase the shelf life of fresh fish [20]. Recently, a new potential approach to prolong the shelf life of fresh products was developed using biopreservation systems [1,23], which consist of the use of natural or controlled microbiota or natural antimicrobials as a way of preserving food and extending its shelf life [1,38]. In particular, the use of lactic acid bacteria (LAB), which have antagonistic properties against spoilage and pathogenic microorganisms and are considered generally recognized as safe (GRAS), has been suggested. Indeed, LAB compete for nutrients and produce metabolites with antimicrobial activities such as lactic and acetic acid, hydrogen peroxide, and peptide bacteriocins [1,20]. Different LAB strains are normally used directly or in combination with other preservative techniques (antimicrobials, sodium alginate) because they can inhibit the activities of a wide spectrum of microorganisms, including spoilers and pathogens in food [39,40].

The aim of this work was to improve the shelf life of farmed sea bass and sea bream using different methods, including vacuum packaging (VP), MAP, and bioprotective culture consisting of *Latilactobacillus sakei*. Microbiological, physicochemical, and sensory quality indices were monitored to confirm the effectiveness of biopreservation on product quality during proper (4 ± 2 °C) or abuse (6 ± 2 °C) temperature storage periods.

## 2. Materials and Methods

Samples consisted of 3 different lots of gutted sea bass (*Dicentrarchus labrax*) and sea bream (*Sparus aurata*). The sea bass weighed approximately 474–578 g and were 35 cm long, and the sea bream weighed 404–440 g and were 25 cm long. Both fish were bred in sea cages by Orada Adriatic d.o.o. in Cres (Split), Croatia, collected, and slaughtered after a bath in water and ice. The postmortem period lasted 24 h on ice, and then the fish were vacuum or modified atmosphere Packaged and stored at different temperatures and times before the analysis. VP (−1.0 bar) and MAP (70% N_2_, <1% O_2_, 30% CO_2_) technologies were chosen according to our experience in the field (data not shown), and applied by an Orved VM53 vacuum machine (Italy). The packaging was a multilayer film consisting of polyethylene terephthalate (PET) with a thickness 12 my, aluminium with a layer thickness 9 my, nylon with a thickness 15 my, and polyethylene (PE) NEUTRO with a thickness 75 my. On different days, three samples were collected and subjected to microbial and physicochemical analysis (moisture, pH, TVB-N, thiobarbituric acid reactive substance (TBARS) assay, rancidity value). Additional samples were subjected to sensorial analysis.

(a)To compare VP and MAP samples stored at 4 ± 2 °C, analyses were performed on days 0, 6, and 12. At each time point and for each packaging condition and fish species, three samples were collected and analysed. Each lot and each type of packaging included 9 sea bass and 9 sea bream samples.(b)To study the microbial and physicochemical development of both fish stored in VP at 6 ± 2 °C (simulating abuse temperatures), analyses were performed on days 0, 3, 6, 9, and 12. At each time point and for each fish species, three samples were collected and analysed. Each lot included 15 sea bass and 15 sea bream samples.(c)To prolong the shelf life of both fish samples stored at 6 ± 2 °C (simulating abuse temperatures) with or without supplementation, a bioprotective starter consisting of Latilactobacillus sakei (LAK-23, Sacco s.r.l., Via Alessandro Manzoni 29/A, 22071 Cadorago, CO, Italy) was used. The strain was selected and isolated from meat products and tested for its genetic and phenotypic characteristics. Then, its use was proposed as a starter for meat fermentation and as a bioprotective agent, being a bacteriocin producer, versus L. monocytogenes and spoilage microorganisms in meat and fish products.

Three samples were collected and analysed on days 0, 7, and 14.

### 2.1. Bioprotective Starter Suspension

The chosen starter was sold freeze-dried in a foil pouch. At the time of use, the starter was thawed, homogenized, and diluted in sterile peptone water (NaCl 0.6%; Peptone, Oxoid, 0.1%, distilled water 1 L). To evaluate its load, dilutions were performed in sterile peptone water, and 0.1 mL of each dilution was inoculated in Petri dishes, to which de Man, Rogosa, and Sharpe (MRS) medium (Oxoid, Italy) was subsequently added by the double layer method. The plates were incubated at 37 °C for 48–72 h, and the grown colonies were counted. Each suspension contained on average approximately 11 log CFU/g. Then, the starter culture was diluted in natural water used to wash the fish before packaging at a level of approximately 7 log CFU/mL.

### 2.2. Bioprotective Starter Inoculum

The three sequential lots of both gutted sea bass and sea bream samples were dipped in three different washing waters and left for 10 min. Each lot included 27 samples for each fish species. The washing waters included the following:(a)Natural water (control), 9 each samples/fish/lot;(b)Natural water supplemented with a bioprotective starter (7 log CFU/mL), 9 samples/fish/lot;(c)Natural water supplemented with a bioprotective starter (7 log CFU/mL) and 0.1% glucose, 9 samples/fish/lot.

Briefly: the chosen *L. sakei* was further examined as a starter to preserve both sea bass and sea bream in VP. *L. sakei* was added to distilled water, which was then used to wash the fish before packaging. In particular, one group of both fish (St) was washed with water supplemented with the starter; another group (StG) was washed with starter supplemented with 0.1% glucose (Oxoid, Italy), and another group (C) was washed with distilled water as a control. After washing and VP, the level of the starter in both fish was approximately 5 log CFU/g. Then, the washed samples were vacuum packed using the abovementioned methodology and stored at 6 ± 2 °C for up to 14 days. On days 0, 7, and 14, three samples were collected and subjected to microbial and physicochemical analysis. Additional washed samples were used at 14 days for sensorial analysis.

### 2.3. Microbiological Analysis

Total viable microbial counts (TVCs) were evaluated on plate count agar (Oxoid, Milan, Italy) incubated at 30 °C for 48–72 h. LAB were counted in MRS medium (Oxoid, Milan, Italy) after incubation at 30 °C for 48 h; total coliforms were counted on violet red bile lactose agar (VRBLA, Oxoid, Milan, Italy) incubated at 37 °C for 24–48 h; *Enterobacteriaceae* were counted in violet bile glucose agar (VRBGA, Oxoid, Milan, Italy) incubated at 37 °C for 48 h; *E. coli* were counted in Coli-Id (bioMèrieux, Marcy-l’Étoile, France) incubated at 37 °C for 48 h; *Pseudomonas* spp. were counted on *Pseudomonas* agar base (Oxoid, Milan, Italy) supplemented with CFC (Oxoid, Milan, Italy) and incubated at 30 °C for 48 h; Enterococci were counted in kanamycin aesculin agar (Oxoid, Milan, Italy) incubated at 37 °C for 48 h; sulfite-reducing Clostridia were quantified in differential reinforced clostridial medium (DRCM, VWR, Radnor, PA, USA) after incubation at 37 °C for 24–48 h in an anaerobic jar, using an anaerobic kit (gas pack anaerobic system, BBL, Becton Dickinson, Franklin Lakes, NJ, USA). *Listeria monocytogenes* and *Salmonella* spp. were investigated according to ISO methods 11290/1 [41] and 6579–1 [42], respectively.

### 2.4. Physico-Chemical Analysis

The pH value was measured using a pH metre (Basic 20, Crison Instruments, Spain), by inserting the probe into 3 different points on each sample. The moisture content was measured by the A.O.A.C. [43] method. The Pearson method [44] was used to evaluate the TVB-N concentration (expressed in milligrams of N/100 g product), and the method described by Ke et al. [45] was used to evaluate the oxidation stability during storage (TBARS; expressed in nanomoles of malonaldehyde/g). Briefly: the total volatile basic nitrogen (TVB-N) was estimated by boiling a mix of distilled water (50 mL) and 10 g product in presence of MgO (25 mL, 2% *w*/*v*). The distillate was collected in a solution of boric acid and titrated with sulfuric acid in the presence of methyl red. Thiobarbituric acid value (TBARS) was determined directly by spectrophotometric quantification of compounds obtained by the distillation of a mix consisting of distilled water (50 mL) and fish product (10 g), acidified with hydrochloric acid (2.5 mL, 4 N) until pH 1.5. Then, 5 mL of the distillate was treated with 5 mL of a solution of thiobarbituric acid (TBA), obtained by mixing TBA (0.2883) in acetic acid (90%) placed in boiled water for 35 min. After cooling, the solution was read at 538 nm. Three analyses were performed at each sampling point.

### 2.5. Sensorial Analysis

Sensorial analyses were performed by 20 nonprofessional and nontrained assessors (10 women and 10 men, representing food technology students aged between 22 and 24 years of age). Nonprofessional assessors were chosen because they represent typical consumers. Each tested sample was packaged in aluminized paper and cooked at 180 °C for 30 min in an oven before the sensorial analysis, which was used to evaluate the following:(a)Comparison between VP and MAP of three sequential lots of gutted sea bass and sea bream samples stored at 4 ± 2 °C at 12 days. Nine samples of each fish species and of each type of packaging were tested;(b)Comparison between VP sea bass and sea bream samples of three sequential lots stored at 6 ± 2 °C for 14 days. The fish samples were labelled as controls if they were not supplemented with a bioprotective starter (samples a), St if they were supplemented with a bioprotective starter (samples b), and StG if they were supplemented with a bioprotective starter and glucose (0.1%) (samples c). Nine samples of each treatment for each lot were tested.

For both tests, sensory analysis was performed using the triangle test methodology, ISO 4120:2004 [46].

After cooking, the products were cooled at 65 °C and used for sensory evaluation.

For test a, the nonprofessional assessors were presented with three products, two of which were identical. The assessors were asked to state which product they believed was the odd one out (*p* < 0.05). The assessors who identified the different samples were asked to indicate their preference. The scoring system used was (VP samples versus MAP samples): 1 (excellent), 2 (good), 3 (sufficient), and 4 (scarce).

For test b, the nonprofessional trained assessors were asked to state which product they believed was a unique sample. The assessors who indicated the presence of two distinct samples were asked to identify the best sample, considering the following parameters: flavour, odour, colour, texture (appearance, surface moisture, and colour), and overall acceptance of the product. The scoring system used was (samples versus samples): 1 (excellent), 2 (good), 3 (sufficient), and 4 (scarce).

### 2.6. Statistical Analysis

Data were analysed using Statistica 7.0 vers. 8 software (Statsoft Inc., Tulsa, OK, USA, 2008). The values of the different parameters were compared by a one-way analysis of variance and the means were then compared using the Tukey’s honest significance test. Differences were considered significant at *p* < 0.05.

## 3. Results and Discussion

### 3.1. Microbial and Physicochemical Evaluation of VP and MAP Gutted Sea Bass and Sea Bream Samples Stored at 4 ± 2 °C

The microbial loads of fresh-gutted, vacuum-packaged, and modified atmosphere-packaged sea bass and sea bream are reported in Table 1 and Table 2. Initially, fish freshness was excellent, and as microbial and TVB-N levels increased, the freshness characteristics gradually diminished with time in both VP and MAP samples. VP seemed to maintain the freshness quality better than MAP. More specifically, the freshness characteristics remained of excellent quality for up to 6 days; however, both VP and MAP fish can also be accepted, considering the level of microbial load reached at 12 days, which was equal or less than 8–9 log CFU/g, representing the microbial concentration required to spoil chilled fish [2,3,4,8]. Conversely, considering the TVB-N value, only the VP samples should be accepted. Indeed, in these VP samples, the TVB-N was always at a level of 35 mg N/100 g, which is the limit of acceptability of fish [47], while the TVB-N of MAP fish exceeded this limit.

Indeed, by consulting the literature, it was determined that the shelf life of fresh fish is based on the storage temperature and atmosphere, the level of initial microbial contamination, and the handling techniques, such as gutting, filleting, and packaging [2,3]. Consequently, in each work, different shelf-life durations are demonstrated. Usually, for fresh whole or gutted sea bass in air or MAP and stored in ice or at 4 °C, the shelf life varies from 8 to 19 days [7,48,49,50]. Our results are not in agreement with those of various authors [2,3,51,52], who found that the shelf life is reduced to 8 days when the fish are commercialized as fillets stored at 2–4 °C under air, or to 12 days in MAP.

The initial concentration of TVC, *Pseudomonas* and *Enterobacteriaceae* (Day 0) was approximately 2 log CFU/g in both the packaging of the fish samples, while other investigated microorganisms (such as *E. coli*, total coliforms, *Clostridium* H_2_S-producing LAB and Enterococci) were not detected (less than the threshold limit of the methods).

*L. monocytogenes* and *Salmonella* spp. were not found in any of the tested samples.

Additionally, TVB-N and TBARS levels were initially acceptable (Table 1).

Over 12 days of storage, different microorganisms, except for *Pseudomonas* and *Clostridium* H_2_S producers, grew. In particular, in vacuum- or modified atmosphere-packaged MAP sea bass, the TVC level exceeded 7 and 8 log CFU/g, respectively. For sea bream, the TVC level was 5.3 log CFU/g in VP and 5.9 log CFU/g in MAP. The main spoilage bacteria, such as Enterobacteriaceae and total coliforms, grew less than in TVC, but both microbial groups exceeded concentrations of 3 and 4 log CFU/g, respectively. VP and MAP and the low level of oxygen (<0.5%) promoted LAB growth; consequently, they reached values between 4.7 and 5.7 log CFU/g (Table 1 and Table 2). In VP for both fish species, the LAB concentration was higher than that for MAP, while the total coliforms were lower. VP and MAP affect not only the growth rate but also the final populations of spoilage bacteria [2,3]. Again, the increase in CO_2_ and the reduction in O_2_, mostly suppressing Gram-negative and favouring Gram-positive microorganisms, increase the shelf life of fresh fish, as Gram-negative microorganisms are the main contributors to spoilage, represented by TVB-N production [2,3,8]. Indeed, the main spoilage bacteria, such as *Pseudomonas* spp. and H_2_S-producing bacteria, grow fast in air, where they become dominant; however, in reduced oxygen environments, their growth is blocked or limited, as demonstrated by our results, and consequently, the shelf life of fresh fishes increases. Most likely, the lower presence of spoilage bacteria justifies the lower TVB-N (*p* < 0.05) concentration and demonstrates the acceptability at 12 days of VP in both fish species. As shown, the TVB-N concentration was 35 mg N/100 g at 12 days in VP for both sea bass and sea bream; this value is considerably acceptable according to Directive 95/149/EEC [47]. Conversely, the TVB-N concentration in MAP fish exceeded the limit, reaching 43.4 mg N/100 g in sea bass and 42.1 mg N/100 g in sea bream samples.

In addition, TBARS values increased (Table 1 and Table 2) and remained at a maximum level of 2.5–2.6 nmol/g at the end of storage (12 days). At 0 days, the TBARS values were 1.5 and 1.6 nmol/g for both fish species, and then they slightly increased (*p* > 0.05), reaching acceptable levels (2.5–2.6 nmol/g); consequently, these values must be accepted. According to several authors [45,53], food products are not rancid when TBARS values are <8 nmol/g of the sample, slightly rancid when TBARS is between 9–20 nmol/g, and rancid and unacceptable when the TBARS is >21 nmol/g.

Indeed, the sensory acceptability of the VP or MAP samples was determined by the triangular test. The jury was composed of 20 nonprofessionally trained evaluators. Fifteen out twenty evaluators perceived only light differences between samples in VP and MAP. Before cooking the sea bass and sea bream samples, the jury agreed in affirming that all samples did not show any white or viscous patinas, slime, discolouration, or browning, or off-flavours or off-odours after cooking; 15 out of 20 evaluators believed the sea bass and sea bream samples in VP maintained the typical odours and flavours of fresh fish better. The scoring system used was (VP versus MAP) one (excellent), two (good), three (sufficient) and four (scarce). Based on this scoring, all samples were acceptable by the 20 evaluators. However, the 15 tasters, who found differences between the two types of packaging, preferred the VP samples with respect to the MAP ones, and the final value score was, respectively, two (good) and three (sufficient). Consequently, they preferred samples of sea bass or sea bream in VP. In any case, the evaluators did not perceive any ammoniacal odour in samples in MAP, although they showed a higher TVB-N concentration than products in VP.

Despite the light differences in the microbial loads and the TVB-N values of MAP samples exceeding 35 mg N/100 g, as proposed by the Directive 95/149/EEC [47], it was concluded that the shelf-life of VP and MAP samples of both the fish was approximatively of about 12 days.

Considering that the best sensorial, physicochemical, and microbial results were obtained for gutted sea bass and sea bream samples in VP stored at 4 ± 2 °C, the next phase of the work was to study the shelf life of gutted sea bass and sea bream in VP stored at 6 ± 2 °C, which simulates the normal abuse temperature of supermarkets or consumer fridges.

### 3.2. Microbial and Physico-Chemical Evaluation of Vacuum-Packaged Sea Bass and Sea Bream Samples Stored at 6 ± 2 °C

The changes in the physicochemical parameters and microbial population of aquacultured gutted sea bream and sea bass in VP during storage at 6 ± 2 °C are shown in Table 3, Table 4, Table 5 and Table 6.

For the sea bass samples, all microorganisms, except *Pseudomonas* spp. and *Clostridium* H_2_S producers, grew during storage. The TVC level of gutted sea bass was 3.7 log CFU/g (Table 3), which was slightly higher than the initial value obtained previously (Table 1). This confirms that the initial contamination depends on different parameters, such as breeding, farming, slaughtering, gutting, filleting, and packaging [2,3,50,51]. Then, TVC grew at 6 days to a level of 6 log CFU/g and after 12 days of storage to over 8.0 log CFU/g. The obtained data agreed with Cakly et al. [9,13], who measured the same concentration in gutted and ungutted sea bass after 14 days of storage at 4 °C. TVC, reaching 8 log CFU/g, exceeded the value of 7 log CFU/g considered to be the maximum level of acceptability for gutted and ungutted freshwater and marine fish [54], and in our experiment, this was reached 2 days before the TVC counts observed by other authors investigating European whole sea bass stored in ice [7,9,13,48,55]. In our experiment, this highest TVC level depended on the abuse temperature of storage (6 ± 2 °C), which is 2–4 °C higher than the storage temperatures (2–4 °C) used by the abovementioned cited authors. Enterobacteriaceae strains and total coliforms were initially measured at levels of 1.4 and 1.6 log CFU/g, respectively, but after 12 days, the counts increased to values higher than 5 log CFU/g.

*E. coli* seemed to have grown, but considering it is a mesophilic strain, it is doubtful that this was real growth but rather growth that depended on the sample, which changed at any analytical time.

Additionally, LAB and Enterococci grew, stimulated by the vacuum (Table 3). In contrast, *Pseudomonas* spp. did not increase solely due to the vacuum, considering that they are closely aerobic. Indeed, *Pseudomonas* spp. are the dominant spoilage microorganisms of chilled stored fish, either air caught or farmed from the warm, temperate waters of the Mediterranean Sea [2,3,7,49,56,57,58], but fish in VP or MAP are dominated by LAB, which are strictly microaerophiles [2,3,59]. In this group of analyses, neither *L. monocytogenes* nor *Salmonella* spp. were found, which demonstrated good hygienic quality and good manufacturing practice applied during the processing of fresh gutted sea bass.

Table 4 shows the physicochemical parameters of the tested gutted sea bass. As shown, the moisture, pH and TBARS values did not change significantly. During storage, the means of the abovementioned parameters seemed to change, but considering the large standard deviation, the change was not significant (*p* > 0.05). In addition, the different means and standard deviations observed could be due to the three analysed samples, which changed at each analytical time. Conversely, the TVB-N concentration changed during storage (*p* < 0.05), which confirmed the effects of microbial growth, as suggested by Hebard et al. [60]. At the beginning of storage, the TVB-N value of the tested gutted sea bass was 12.9 ± 0.3 mg N/100 g and then increased and reached a value of approximately 39 ± 1.3 mg N/100 g (Table 4). This value indicates that spoilage had just started at 12 days. Indeed, at 6 days of storage, the TVB-N value was approximately 31.5 ± 1.3 mg N/100 g; this value is considered acceptable according to the limit proposed by EC/1995 [52], which is 35 mg N/100 g. The initial TVB-N concentration is typically between 5 and 20 mg N/100 g [9,13], but at the end of storage, it was over 30–35 N/100 g, which is the concentration that is generally regarded as the limit of acceptability for ice-stored cold water fish [10,61].

However, despite the microbial level (8 log CFU/g) and TVB-N values (39 mg N/100 g), the gutted sea bass samples must be accepted, considering that there was no unacceptable odour and that Cakly et al. [9,13] suggested the acceptability of aquacultured sea bass stored in ice, which presented a TVC value above 8 log CFU/g and TVB-N and TBARS values of approximately 50.13 ± 0.25 mg N/100 g, and 2.66 ± 0.06 mg malonaldehyde/kg, respectively.

During the storage of sea bream samples, all the microorganism groups, except *Clostridium* H_2_S and Enterococci, grew. At the beginning the TVC concentration was approximately 2.3 log CFU/g (Table 5). Then, it grew, and at 12 days reached 5.5 ± 1.9 log CFU/g. This concentration was similar to data of Cakly et al. [9,13], who measured the same values in gutted and ungutted sea bream after 7 days of storage at 4 °C. Consequently, the level of TVC indicates the fish can be largely acceptable, considering that the final TVC did not exceed 7 log CFU/g, as requested for gutted and ungutted freshwater and marine fish [54].

This adequate TVC value confirms the application of excellent production processes [2,3,50,51], and although it was obtained at abuse temperatures (6 ± 2 °C), it was lower than that obtained by different authors for European whole sea bream stored in ice [2,3,9,13,48,50,51]. Additionally, the concentrations of Enterobacteriaceae and total coliforms were initially low, 2.1 ± 0.3 log CFU/g and less than 10 CFU/g, respectively. During 12 days of storage, both microbial groups grew to 4.9 ± 0.4 log CFU/g and 4.5 ± 0.8 log CFU/g, respectively. *E. coli* did not grow, and at each analysis time, the difference was not significant (*p* > 0.05). Additionally, LAB grew and reached 5.5 ± 0.4 log CFU/g, considering that they are microaerophilic (Table 3). Conversely, *Pseudomonas* spp. demonstrated soft growth, dependent on the residual oxygen in the VP, because *Pseudomonas* spp. are strictly aerobic, but the averages at any time were not significantly different (*p* > 0.05). In sea bream, a higher microbial concentration was represented by LAB, and are strictly microaerophiles [2,3,59]. Additionally, neither *L. monocytogenes* nor *Salmonella* spp. were found in sea bream samples.

The physicochemical parameters of the tested gutted sea bream are shown in Table 6. The moisture, pH, and TBARS values did not significantly change (*p* > 0.05). Only the TVB-N changed at any time during the analysis (*p* < 0.05), confirming the effects of microbial growth suggested by different authors [2,3,9,13,55,60]. At the beginning of storage, the TVB-N values of the tested gutted sea bream were similar to those of sea bass and were determined to be 12.3 ± 0.2 mg N/100 g. Then, the TVB-N value increased according to the time of storage and reached a value of approximately 35.0 ± 1.2 mg N/100 g (Table 6) at 12 days of storage. This value must be considered largely acceptable according to the limit proposed by EEC/1995 [52] and for ice-stored cold water fish [10,61], which is 35 mg N/100 g. The final TVB-N concentration in sea bream samples was lower than that in sea bass samples because the former contained a lower concentration of spoilage microorganisms. The levels of TVC and Enterobacteriaceae in sea bream were 2.5 and 1 log CFU/g lower, respectively, than those in sea bass. Therefore, it could be demonstrated that the presence of lower spoilage microorganism concentrations corresponds to lower TVB-N concentrations. Microbial and physicochemical data demonstrated that the sea bream tested must be accepted, considering that there was no unacceptable odour and that they presented a TVB-N of less than 8 log CFU/g, a TVB-N of less than 50.13 ± 0.25 mg N/100 g and a TBARS of less than 2.66 ± 0.06 mg malonaldehyde/kg [9,13].

Finally, considering the TVC, TVB-N, and TBARS values, it seems that both the VP gutted fish can be accepted until 12 days of storage at 6 ± 2 °C and, consequently, this time can represent the limit of their shelf-life.

Considering their economic value and the growing interest of consumers in their nutritional aspects, the next aim was to prolong their shelf life until 14 days. Fresh fish are rapidly susceptible to spoilage due to microbiological and biochemical degradation [1,17], and, to extend their shelf life, different preservative technologies are used, such as heat processing, chemical preservatives, MAP, and refrigeration [1]. These technologies are extensively used, but they do not completely control spoilage bacteria. In particular, some technologies, such as heat processing and antimicrobial compounds, cannot be used to preserve fresh fish. Heat processing changes the texture of fish, which becomes processed food, and synthetic preservatives are not acceptable by consumers, who increasingly demand high-quality, but minimally processed, seafood [62]. Therefore, the abovementioned technologies cannot be used to preserve fish. The use of bioprotective methods is a new, modern, and promising method largely used in other food fields to obtain good results against spoilage and pathogenic microorganisms without changing the texture, flavour, or odour of the product [23,63,64]. Among LAB, *Latilactobacillus sakei* is frequently used in bioprotective technology [65,66]. In particular, LAK-23, a commercialized bioprotective starter culture based on *L.*
*sakei*, was chosen to try to achieve our objective, considering that LAB originally isolated from certain food products are the best starter cultures for these same products, because they would be more competitive than LAB from other sources [23,67]. Starter cultures and LAB, in particular, are considered as GRAS by the Food and Drug Administration [68]. This status may be based either on a history of safe use in food prior to 1958 or on scientific procedures, which require the same quantity and quality of evidence as would be required to obtain food additive regulations. In Europe, starter cultures are granted Qualified Presumption of Safety (QPS) status if reasonable evidence is provided. A safety assessment can be made based on four pillars: taxonomic identification, body of knowledge, possible pathogenicity (‘safety concerns’), and end use [69]. The body of knowledge is one of the pillars of the QPS evaluation and is investigated based on the scientific literature [70]. QPS provides a safety status for microorganisms intentionally used in the food and feed chain, certifying that they do not pose a risk to human and animal health [69,70]. Consequently, *L. sakei* is traditionally and largely used as a starter to promote food ripening, and as a bioprotective agent against pathogenic and spoilage microorganisms.

The data regarding the different washing treatments are shown in Table 7 and Table 8. During storage at 6 ± 2 °C, the starter consisting of *Latilactobacillus sakei* grew until the end of the experiment (14 days) and reached values over 6 log CFU/g; conversely, in the control samples, the level of autochthonous LAB was always less than 5.5 log CFU/g (Table 7 and Table 8). LAB growth inhibited spoilage microorganisms such as total coliforms and *Enterobacteriaceae*, considering that *Pseudomonas* growth was blocked by LAB and, above all, by VP. Indeed, at the end of the storage, the *Enterobacteriaceae* concentrations in the sea bass samples washed with starter (St) and starter added with sugar (StG) were lower than in the samples washed with water (C), and were 4.4 ± 0.1 and 3.3 ± 0.2 CFU/g (*p* < 0.05), respectively. Conversely, in the C samples, they were 4.9 ± 0.3 CFU/g. Different concentrations were also present at level of total coliforms (*p* < 0.05). Indeed in C., St, and StG samples the total coliforms reached values of 5.0 ± 0.3, 4.1 ± 0.3, and 3.1 ± 0.2 CFU/g, respectively. Similar behaviour could be observed in sea bream samples. Indeed, the *Enterobacteriaceae* concentrations in samples washed with starter (St) and starter added with sugar (StG) were lower than in the samples washed with water (C), and were 4.3 ± 0.1 and 3.4 ± 0.2 CFU/g (*p* < 0.05), respectively. Conversely, in the C samples, they were 4.9 ± 0.3 CFU/g. The different concentrations were also present at the level of total coliforms (*p* < 0.05). Indeed in C., St, and StG samples the total coliforms reached values of 4.4 ± 0.3, 4.0 ± 0.3, and 3.0 ± 0.2 CFU/g, respectively. The reduced growth of both the total coliforms and *Enterobacteriaceae* depended on the added LAB starters, which grew over 6 log CFU/g. Indeed, in sea bass and sea bream C samples, the LAB reached 5.3 ± 0.2 and 5.3 ± 0.1 CFU/g, while in St they were 6.5 ± 0.2 and 6.0 ± 0.1 CFU/g and in StG they were 7.2 ± 0.2, and 6.9 ± 0.5 CFU/g, respectively.

In addition, the TVC level was similar in all samples independent of the starter, and no significant differences were observed among the samples (*p* > 0.05), as found by Bassi et al. [23]. Finally, in these groups of fish, neither *L. monocytogenes* nor *Salmonella* spp. was ever found. The activity of the starters was confirmed by the change in pH and TVB-N level. Indeed, the final pH was approximately 6.08 (St) and 6.04 (StG) in sea bass-inoculated samples and 6.04 (St) and 6.02 (StG) in sea bream-inoculated samples, while in the controls, the final pH was 6.11 in both fish species. These data do not agree with those of other authors, who found that in vacuum-packed sea bass, the pH decreased to 5.6 units [23]. In StG samples, a higher pH decrease was expected because of the added sugar. This can be explained by the limited final LAB loads (less than 7.5 CFU/g). In each case, its value was less than that in the control samples, where the pH decrease was very limited, given the small level of glucose initially present in the fish flesh [8].

Again, the TVB-N value increased in all samples. At 14 days of storage, the TVB-N concentration of the StG samples was approximately 30.2 and 31.2 mg N/100 g and that of the St samples was approximately 37.2 and 38.3 mg N/100 g in sea bass and sea bream, respectively. Conversely, in the C samples for both fish species, the level of TVB-N was always greater than 40 mg N/100 g (Table 7 and Table 8).

This lower TVB-N value in the StG and St samples depends on the reduced activity of *Enterobacteriaceae*, as previously demonstrated by Gram and Huss [8]. Indeed, *Enterobacteriaceae* and, consequently, total coliforms, are recognized to be responsible for TVB-N and trimethylamine production [8,23]. Therefore, the starter LAB suppressed the spoiling bacteria, yielding a reduction in the TVB-N concentration. This is in agreement with data on the LAB inoculation effect [8,23,65,71]. The abovementioned authors noticed that the use of starter cultures with antimicrobial properties against *Listeria* sp. and psychotropic bacteria could reduce the risk of biogenic amine and, consequently, TVB-N formation, whose production in vacuum-packed fishes depends on psychotropic bacteria that proliferate slowly and dominate the mesophilic bacterial load, because low temperatures favour their growth [8,30,59,72,73,74,75]. Finally, the TBARS levels of all the tested samples (C, St, StG) always remained less than or equal to 2.2 nmol/g, demonstrating that VP protects both fish from rancidity (Table 7 and Table 8). Additionally, in this case, all the samples can be acceptable given the TBARS values, as suggested by Cakly et al. [9,13].

Based on the physicochemical results, it can be concluded that the use of starter culture can prolong the shelf life of sea bass and sea bream in VP until 14 days of storage at 6 ± 2 °C, a temperature that is considered typical of supermarkets and consumer fridges. The TVC, Enterobacteriaceae, total coliforms, and TVB-N concentrations of the StG-inoculated samples met the limit proposed by the ICMSF [54] and EEC/1995 [52], and consequently, they must be largely acceptable until 14 days at 6 ± 2 °C. Additionally, the samples treated with only the starter can be accepted, despite the level of TVB-N exceeding the limit proposed by EEC/1995 [52].

### 3.3. Sensorial Analysis

In addition to food preservation, sensory characteristics cannot be neglected as the main factors responsible for product acceptance [1]. Indeed, the samples developed in this study were also judged by nontrained nonprofessional evaluators; therefore, it was decided to show only the overall quality attributes, because the other sensory descriptors demonstrated a similar trend. The sensory assessment was performed on day 14 of the storage period, because, on this day, the products could be accepted, considering the microbial and TVB-N level, and the results obtained by a triangular test are shown in Table 9. The obtained results demonstrated that all nonprofessional evaluators identified the samples based on the three treatments (Table 9) and that there were no great differences in the samples.

The small difference in pH between samples washed with St, STG, and C was not valued significantly by the panellists. Indeed, the panel identified a slightly acidic, nondisturbing taste in the St and StG samples.

Finally, the three treatments were differentiated by the score attributed to the investigated parameters, such as flavour, odour, colour, texture (appearance, surface moisture, and colour), and overall acceptance of the product. The scoring system used was (samples versus samples) one (excellent), two (good), three (sufficient), and four (scarce). Based on this scoring, all samples were acceptable; in particular, the 20 evaluators preferred the following in descending order of acceptability: StG, St, and C. Therefore, it was proposed to use bioprotective starters diluted in water supplemented with glucose to prolong the shelf life until 14 days at 6 °C for either fresh sea bass or fresh sea bream.

## 4. Conclusions

Fish meat is very perishable because of indigenous and microbial enzymes, which determine spoilage and shelf life. The deterioration processes depends on different parameters and, in particular, by the type of packaging and by the storage conditions. In this paper, different technologies have been used in order to prolong the shelf-life of fresh sea bass and sea bream. Vacuum and modified atmosphere packaging were first compared in order to choose the best packaging for both the fish. Independently of the type of packaging, data showed light differences between the fish samples during proper refrigeration (4 ± 2 °C) and a similar shelf-life of about 12 days. However, either for TVB-N values or for sensorial analysis, the VP was considered the best packaging. Indeed, based on the applied scoring, the 15 out 20 tasters, who found difference between the two type of packaging, preferred the VP samples with respect to MAP ones, and the final value score was respectively two (good) and three (sufficient).

The shelf-life of 12 days was also confirmed by the evaluation of microbiological, physicochemical (TVB-N), and sensory quality indices in VP fish stored at abuse temperature (6 ± 2 °C, simulating supermarkets and consumer fridges) during the storage period.

However, to prolong the shelf life of both the fish, different methods occurred. In particular, a method was employed that washed the gutted sea bass and sea bream in water added with or without dextrose (0.1%), and inoculated them with bioprotective starter (7 log CFU/mL). After washing the samples, they were subjected to VP and stored at 6 ± 2 °C. The bioprotective starter permitted a reduction in growth of spoilage microorganisms and the increasing of the TVB-N concentration, which in both fish was less than 35 mg N/100 g product. Consequently, the shelf-life of both fish was about 14 days. In addition, nonprofessional and untrained evaluators confirmed the acceptability of the inoculated samples by sensorial analysis. Indeed, they considered the fish treated with StG excellent, and the ones treated with St good.

## Figures and Tables

**Table 1 biology-11-00217-t001:** Development of microorganisms, TVB-N and TBARS in gutted sea bass packaged under vacuum or in MAP and stored at 4 ± 2 °C.

Microorganisms	Time (Days)					
	T0		T6		T12	
	VP	MAP	VP	MAP	VP	MAP
Total viable count	2.0 ± 0.2 a	2.0 ± 0.2 a	5.3 ± 0.2 a	7.9 ± 0.5 b	7.9 ± 0.5 a	8.7 ± 0.9 b
*Enterobacteriaceae*	2.1 ± 0.2 a	2.1 ± 0.2 a	3.4 ± 0.4 a	3.6 ± 0.1 a	4.0± 0.7 a	4.5 ± 0.8 a
*Pseudomonas*	2.3 ± 0.2 a	2.3 ± 0.2 a	2.3 ± 1.2 a	2.6 ± 0.6 a	2.1 ± 0.6 a	2.4 ± 0.5 a
*E. coli*	<10 a	<10 a	2.0 ± 0.3 a	2.0 ± 0.2 a	2.0 ± 0.5 a	2.0 ± 0.6 a
Total coliforms	<10 a	<10 a	2.5 ± 0.4 a	2.3 ± 0.3 a	4.0 ± 0.3 a	4.8 ± 0.1 b
*Clostridium* H_2_S+	<10	<10	<10	<10	<10	<10
Lactic acid bacteria	<10 a	<10 a	2.2 ± 0.4 a	2.2 ± 0.3 a	5.7 ± 0.2 a	5.0 ± 0.3 b
Enterococci	<10 a	<10 a	2.0 ± 0.1 a	2.0 ± 0.2 a	2.6 ± 0.5 a	2.3 ± 0.3 a
TVB-N	13.0 ± 0.2 a	13.5 ± 0.5 a	19.9 ± 0.5 a	20.1 ± 0.3 a	35.2 ± 0.1 a	43.4 ± 0.2 b
TBARS	1.5 ± 0.2 a	1.5 ± 0.1 a	2.7 ± 0.3 a	2.4 ± 0.3 a	2.5 ± 0.3 a	2.6 ± 0.2 a

Legend: Data represent the means ± standard deviations of the total samples; Mean with the same letters within each line (following the values), regardless of packaging method and storage time are not significantly differently (*p* < 0.05). Analyses were conducted in triplicate on three different samples per each sampling point. Data log CFU/g; <10 CFU/g; TVB-N—Total volatile basic nitrogen mg N/100 g; TBARS: nmol malonaldehyde/g.

**Table 2 biology-11-00217-t002:** Development of microorganisms, TVB-N and TBARS in sea bream packaged under vacuum or in MAP and stored at 4 ± 2 °C.

Microorganisms	Time (Days)					
	T0		T6		T12	
	VP	MAP	VP	MAP	VP	MAP
Total viable count	2.3 ± 0.1 a	2.3 ± 0.2 a	4.5 ± 1.5 a	5.4 ± 0.2 a	5.3 ± 0.3 a	5.9 ± 0.2 b
*Enterobacteriaceae*	2.1 ± 0.3 a	2.0 ± 0.1 a	2.6 ± 0.3 a	2.3 ± 0.1 a	3.9 ± 0.4 a	4.7 ± 0.3 b
*Pseudomonas*	2.4 ± 0.1 a	2.3 ± 0.2 a	2.8 ± 1.6 a	2.7 ± 0.3 a	2.0 ± 0.2 a	2.4 ± 0.6 a
*E. coli*	<10 a	<10 a	2.1 ± 0.1 a	2.2 ± 0.3 a	2.1 ± 0.3 a	2.4 ± 0.2 a
Total coliforms	<10 a	<10 a	1.9 ± 0.8 a	2.0 ± 0.7 a	3.5 ± 0.4 a	4.7 ± 0.5 b
*Clostridium* H_2_S+	<10	<10	<10	<10	<10	<10
Lactic acid bacteria	<10 a	<10 a	2.4 ± 0.7 a	2.0 ± 0.1 a	5.5 ± 0.4 a	4.7 ± 0.2 b
Enterococci	<10 a	<10 a	2.0 ± 0.1 a	2.0 ± 0.2 a	2.0 ± 0.1 a	2.9 ± 0.9 a
TVB-N	12.9 ± 0.5 a	12.7 ± 0.3 a	21.9 ± 0.1 a	23.5 ± 1.2 a	35.0 ± 0.2 a	42.1± 0.3 b
TBARS	1.6 ± 0.2 a	1.5 ± 0.2 a	2.5 ± 0.2 a	2.6 ± 0.3 a	2.5 ± 0.3 a	2.6 ± 0.2 a

Legend: Data represent the means ± standard deviations of the total samples; Mean with the same letters within each line (following the values), regardless of packaging method and storage time are not significantly differently (*p* < 0.05). Analyses were conducted in triplicate on three different samples per each sampling point. Data log CFU/g; <10 CFU/g; TVB-N—Total volatile basic nitrogen mg N/100 g; TBARS: nmol malonaldehyde/g.

**Table 3 biology-11-00217-t003:** Development of microorganisms in gutted sea bass packaged under vacuum and stored at 6 ± 2 °C.

Microorganisms	Time (Days)				
	0	3	6	9	12
Total viable count	3.7 ± 1.2 a	5.7 ± 0.4 b	6.0 ± 0.2 b	7.4 ± 0.1 c	8.0 ± 0.4 d
*Enterobacteriaceae*	1.4 ± 0.1 a	3.5 ± 0.3 b	3.8 ± 0.3 b	4.3 ± 0.6 b	5.8 ± 0.1 c
*Pseudomonas* spp.	2.4 ± 0.7 a	2.0 ± 0.2 a	2.0 ± 0.3 a	2.1 ± 0.1 a	2.2 ± 0.1 a
*E. coli*	<10 a	2.7 ± 0.2 b	2.9 ± 0.1 b	<10 a	3.6 ± 0.6 c
Total Coliforms	1.6 ± 0.1 a	3.5 ± 0.1 b	3.3 ± 0.2 b	3.5 ± 0.1 b	5.1 ± 0.2 c
*Clostridium* H_2_S+	<10	<10	<10	<10	<10
Lactic acid bacteria	<10 a	3.7 ± 0.4 b	4.7 ± 0.2 c	6.0 ± 0.3 d	6.1 ± 0.7 d
Enterococci	<10 a	<10 a	<10 a	2.9 ± 0.3 b	3.4 ± 0.7 b

Legend: Data represent the means ± standard deviations of the total samples; Mean with the same letters within each line (following the values), regardless of packaging method and storage time are not significantly differently (*p* < 0.05). Analyses were conducted in triplicate on three different samples per each sampling point. Data log CFU/g; <10 CFU/g.

**Table 4 biology-11-00217-t004:** Physicochemical values of gutted sea bass packaged under vacuum and stored at 6 ± 2 °C.

Parameter			Time (Days)		
	**0**	**3**	**6**	**9**	**12**
Moisture	79.5 ± 0.3 a	77.6 ± 0.9 b	76.3 ± 0.9 b	77.2 ± 2.0 b	76.6 ± 0.8 b
pH	6.16 ± 0.03 a	6.03 ± 0.09 a	6.06 ± 0.07 a	5.91 ± 0.01 a	6.03 ± 0.04 a
TVB-N	12.9 ± 0.3 a	11.0 ± 3.5 a	21.0 ± 0.9 b	31.5 ± 1.3 c	39.0 ± 1.2 d
TBARS	1.6 ± 1.2 a	2.4 ± 1.2 a	2.8 ± 0.5 a	2.4 ± 0.6 a	2.6 ± 0.3 a

Legend: Moisture %, TVB-N—Total volatile basic nitrogen mg N/100 g; TBARS: nmol malonaldehyde/g. Data represent the means ± standard deviations of the total samples; Mean with the same letters within a lanes (following the values), considering each single parameter regardless of the times, are not significantly differently (*p* < 0.05). Analyses were conducted in triplicate on three different samples per each sampling point.

**Table 5 biology-11-00217-t005:** Development of microorganisms in sea bream packaged under vacuum and stored at 6 ± 2 °C.

Microorganisms	Time (Days)				
	0	3	6	9	12
Total viable count	2.3 ± 0.1 a	2.3 ± 0.2 a	4.5 ± 1.5 b	5.4 ± 1.2 b	5.5 ± 1.9 b
*Enterobacteriaceae*	2.1 ± 0.3 a	2.0 ± 0.1 a	2.6 ± 0.3 b	2.3 ± 0.1 b	4.9 ± 0.4 c
*Pseudomonas* spp.	2.2 ± 0.3 a	2.0 ± 0.4 a	2.0 ± 0.5 a	2.1 ± 0.2 a	2.5 ± 0.3 a
*E. coli*	<10 a	<10 a	2.1 ± 0.1 b	2.2 ± 0.3 b	2.1 ± 1.1 b
Total coliforms	<10 a	<10 a	1.9 ± 0.8 b	2.0 ± 0.9 b	4.5 ± 0.8 c
*Clostridium* H_2_S+	<10	<10	<10	<10	<10
Lactic acid bacteria	<10 a	<10 a	2.4 ± 0.7 b	2.0 ± 0.1 b	5.5 ± 0.4 c
Enterococci	2.0 ± 0.1 a	2.0 ± 0.2 a	2.0 ± 0.2 a	2.1 ± 0.2 a	2.0 ± 0.1 a

Legend: Data represent the means ± standard deviations of the total samples; Mean with the same letters within a lanes (following the values), considering each single parameter regardless of the times, are not significantly differently (*p* < 0.05). Analyses were conducted in triplicate on three different samples per each sampling point. Data log CFU/g; <10 CFU/g.

**Table 6 biology-11-00217-t006:** Physicochemical values of gutted sea bream packaged under vacuum and stored at 6 ± 2 °C.

Parameter	Time (Days)				
	0	3	6	9	12
Moisture	75.3 ± 0.1 a	75.6 ± 0.3 a	76.1 ± 0.2 b	76.2 ± 0.3 b	76.0 ± 0.2 b
pH	6.1 ± 0.1 a	6.0 ± 0.1 a	6.1 ± 0.1 a	5.9 ± 0.1 a	6.0 ± 0.1 a
TVB-N	12.3 ± 0.2 a	11.3 ± 1.5 a	22.0 ± 0.3 b	33.2 ± 0.3 c	35.0 ± 1.2 d
TBARS	1.2 ± 0.8 a	2.2 ± 0.9 a	2.4 ± 0.3 a	2.6 ± 0.3 a	2.7 ± 0.2 a

Legend: Moisture %, TVB-N—Total volatile basic nitrogen mg N/100 g; TBARS: nmol malonaldehyde/g. Data represent the means ± standard deviations of the total samples; Mean with the same letters within a lanes (following the values), considering each single parameter regardless of the times, are not significantly differently (*p* < 0.05). Analyses were conducted in triplicate on three different samples per each sampling point.

**Table 7 biology-11-00217-t007:** Development of microorganisms, TVB-N, and TBARS in VP packaged sea bass added with or without bioprotective cultures and glucose (0.1%) and stored at 6 ± 2 °C.

Microorganisms	Time (Days)								
	T0			T7			T14		
	C	St	StG	C	St	StG	C	St	StG
Total viable count	2.0 ± 0.2 a	2.0 ± 0.1 a	2.1 ± 0.3 a	3.3 ± 0.2 b	3.1 ± 0.3 b	4.1 ± 0.1 c	6.0 ± 0.1 d	6.4 ± 0.3 d	6.3 ± 0.2 d
*Pseudomonas* spp.	2.3 ± 0.2 a	2.3 ± 0.1 a	2.0 ± 0.2 a	2.3 ± 0.5 a	2.3 ± 0.5 a	2.4 ± 0.1 a	2.1 ± 0.3 a	2.1 ± 0.2 a	2.2 ± 0.1 a
Lactic acid bacteria	2.0 ± 0.1 a	5.0 ± 0.3 b	5.0 ± 0.3 b	2.8 ± 0.4 a	6.2 ± 0.2 c	7.0 ± 0.5 d	5.3 ± 0.2 b	6.5 ± 0.2 c	7.2 ± 0.5 d
Enterococci	<10^2^ a	<10^2^ a	<10^2^ a	2.0 ± 0.1 a	2.0 ± 0.1 a	2.0 ± 0.1 a	2.6 ± 0.5 a	2.6 ± 0.5 a	2.3 ± 0.3 a
Total coliforms	<10 a	<10 a	<10 a	3.0 ± 0.2 a	2.5 ± 0.4 a	2.2 ± 0.5 b	5.0 ± 0.3 c	4.1 ± 0.3 d	3.1 ± 0.2 b
*E. coli*	<10 a	<10 a	<10 a	2.3 ± 0.3 b	2.1 ± 0.3 b	2.2 ± 0.1 b	2.5 ± 0.3 b	2.4 ± 0.3 b	2.3 ± 0.2 b
*Enterobacteriaceae*	2.1 ± 0.2 a	2.3 ± 0.1 a	2.2 ± 0.2 a	3.4 ± 0.4 a	2.9 ± 0.1 a	2.5 ± 0.3 a	4.9 ± 0.3 b	4.4 ± 0.1 b	3.3 ± 0.2 a
*Clostridium* H_2_S+	<10	<10	<10	<10	<10	<10	<10	<10	<10
pH	6.0 ± 0.1 a	6.0 ± 0.1 a	6.0 ± 0.1 a	6.0 ± 0.2 a	6.0 ± 0.1 a	6.0 ± 0.3 a	6.1 ± 0.2 a	6.0 ± 0.2 a	6.0 ± 0.1 a
TVB-N	12.7 ± 0.1 a	12.9 ± 0.1 a	12.9 ± 0.3 a	22.5 ± 1.5 b	21.5 ± 1.5 b	19.0 ± 1.2 b	42.2 ± 0.2 c	37.2 ± 1.2 d	30.2 ± 0.3 e
TBARS	1.7 ± 0.2 a	1.5 ± 0.1 a	1.7 ± 0.3 a	2.0 ± 0.3 a	2.0 ± 0.1 a	2.1 ± 0.2 a	2.2 ± 0.3 a	2.3 ± 0.1 a	2.2 ± 0.2 a

Legend: C: Control: without bioprotective starter; St: with bioprotective starter, and StG: with bioprotective starter and glucose (0.1%) added. Data represent the means ± standard deviations of the total samples; Mean with the same letters within a lanes (following the values) are not significantly different (*p* < 0.05). Analyses were conducted in triplicate on three different samples per each sampling point. Data log CFU/g; <10–10^2^ CFU/g; TVB-N—Total volatile basic nitrogen mg N/100 g; TBARS: nmol malonaldehyde/g.

**Table 8 biology-11-00217-t008:** Development of microorganisms and TVB-N, and TBARS in VP packaged sea bream added with or without bioprotective cultures and glucose (0.1%) and stored at 6 ± 2 °C.

Microorganisms	Time (Days)								
	T0			T7			T14		
	C	St	StG	C	St	StG	C	St	StG
Total viable count	2.3 ± 0.2 a	2.3 ± 0.1 a	2.3 ± 0.3 a	4.3 ± 0.2 b	4.1 ± 0.3 b	4.0 ± 0.1 b	6.9 ± 0.3 c	6.2 ± 0.5 c	6.1 ± 0.4 c
*Pseudomonas* spp.	2.3 ± 0.2 a	2.3 ± 0.1 a	2.0 ± 0.2 a	2.8 ± 1.2 a	2.3 ± 0.3 a	2.0 ± 0.2 a	2.3 ± 0.5 a	2.1 ± 0.2 a	2.1 ± 0.1 a
Lactic acid bacteria	2.0 ± 0.1 a	5.0 ± 0.3 b	5.0 ± 0.5 b	2.8 ± 0.6 a	5.9 ± 0.2 b	6.8 ± 0.3 c	5.3 ± 0.1 d	6.0 ± 0.1 b	6.9 ± 0.5 c
Enterococci	<10^2^ a	<10^2^ a	<10^2^ a	2.0 ± 0.2 a	2.0 ± 0.3 a	2.0 ± 0.1 a	2.5 ± 0.3 b	2.6 ± 0.3 b	2.0 ± 0.5 ab
Total coliforms	<10 a	<10 a	<10 a	2.2 ± 0.4 b	2.3 ± 0.3 b	2.1 ± 0.3 b	4.4 ± 0.3 c	4.0. ± 0.3 c	3.0 ± 0.2 d
*E. coli*	<10 a	<10 a	<10 a	2.3 ± 0.3 b	2.1 ± 0.3 b	2.2 ± 0.1 b	2.1 ± 0.3 b	2.1 ± 0.1 b	2.1 ± 0.2 b
*Enterobacteriaceae*	2.2 ± 0.3 a	2.2 ± 0.1 a	2.2 ± 0.1 a	2.5 ± 0.2 a	2.3 ± 0.6 a	2.2 ± 0.3 a	4.9 ± 0.3 b	4.3 ± 0.1 c	3.4 ± 0.2 d
*Clostridium* H_2_S+	<10	<10	<10	<10	<10	<10	<10	<10	<10
pH	6.0 ± 0.1 a	6.0 ± 0.1 a	6.0 ± 0.1 a	6.0 ± 0.1 a	6.0 ± 0.3 a	6.0 ± 0.3 a	6.1± 0.1 a	6.0 ± 0.3 a	6.0 ± 0.2 a
TVB-N	12.3 ± 0.2 a	12.2 ± 0.3 a	12.2 ± 0.1 a	22.9 ± 0.5 b	21.9 ± 0.8 b	19.0 ± 0.2 c	42.5 ± 1.2 d	38.2 ± 0.8 e	31.2 ± 0.2 f
TBARS	1.5 ± 0.2 a	1.5 ± 0.1 a	1.6 ± 0.2 a	2.0 ± 0.1 b	2.0 ± 0.2 b	2.1 ± 0.1 b	2.2 ± 0.1 b	2.1 ± 0.2 b	2.2 ± 0.1 b

Legend: C: Control: without bioprotective starter; St: with bioprotective starter, and StG: with bioprotective starter and glucose (0.1%) added. Data represent the means ± standard deviations of the total samples; Mean with the same letters within a lanes (following the values). are not significantly different (*p* < 0.05). Analyses were conducted in triplicate on three different samples per each sampling point. Data log CFU/g; <10–10^2^ CFU/g; TVB-N—Total volatile basic nitrogen mg N/100 g; TBARS: nmol malonaldehyde/g.

**Table 9 biology-11-00217-t009:** Sensorial evaluation by not professional trained panellists.

Fishes Samples		Difference	Final Values Score *
	C versus St	+20/20	3/1
Sea bass	C versus StG	+20/20	3/1
	St versus StG	+20/20	2/1
	C versus St	+20/20	3/1
Sea bream	C versus StG	+20/20	3/1
	St versus StG	+20/20	2/1

Legend: + n. positive assessments/total assessments; C not inoculated samples with. bioprotective starter; St samples inoculated with bioprotective starter; StG samples inoculated with bioprotective starter and added with dextrose (0.1%). * Scores (samples versus samples) 1 (excellent). 2 (good). 3 (sufficient).

## Data Availability

The data in this study are readily available upon reasonable request to the corresponding author.

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
