# Peer review of "Evaluation of Different Techniques, including Modified Atmosphere, under Vacuum Packaging, Washing, and Latilactobacillus sakei as a Bioprotective Agent, to Increase the Shelf-Life of Fresh Gutted Sea Bass (Dicentrarchus labrax) and Sea Bream (Sparus aurata) Stored at 6 ± 2 °C"

_biology, 2022, doi:10.3390/biology11020217_

Round 1

Reviewer 1 Report

The manuscript has been performed. However, some basic aspects remain and give me many doubts about convenience for accepting this manuscript in a journal (i.e., Q1 rank) as the present one.

Two major aspects:

As previously mentioned, the new approach is presented in Tables 7-8. Previous tables provide a comparison that does not include new information, and whose results can be easily expected. Additionally, Tables 7-8 only provide two sampling points (i.e. 7 and 14), so that detailed evolution of quality loss in all three packaging methods is not provided.

Another aspect that is not clearly explained is the sampling procedure. In some cases, three analyses are mentioned. In others, triplicates are mentioned. Both explanations are very different. This has to be expressed uniformly.  

Minor aspects:

The word “increase” is still present in the title. What is to be increased ?

Minor details on physicochemical methods are still not provided. Contrary, authors provide extended information on other kinds of quality determinations.

Author Response

See the upload below

Reviewer 2 Report

The paper contains information useful in a field of methods of extending the life of fish. Overall, the topic discussed is interesting, however the article contains many inaccuracies that need to be clarified and corrected. According to my opinion, the title does not reflect the manuscript content and does not correspond to the purpose of the work. The goal set by the authors covers more factors than the title suggests. The title should be changed to include packing system (VP, MAP) as a factor to be considered.

Abstract reflects manuscript content. All relevant parts of the manuscript are summarized. Keywords are well-chosen, as well. The introduction generally justifies the choice of the topic of the work, although the authors do not explain why was a culture consisting of Lactobacillus sakei selected?

Some data is missing in the Material and methods section: it is unclear how many fish were used within each lot, how many fish were used in the experiment. In chapter 2.5, the authors write that the sensory analysis covered samples that differed by the packaging system, although the results were not presented. It is not clear what the temperature of the fish was during the sensory analysis - it was carried out immediately after cooking (hot) or after cooling.

Tables 1 and 2 are not well organized - TBARS are not microorganisms. The units in which the quantities were expressed were also not given.

Table 7 – Clostridi??

Tables 7 and 8 - Why standard deviations for PH were not indicted?

Conclusions are supported by the data, however there is no conclusion on the results of the sensory evaluation.

Author Response

See the upload below

Reviewer 3 Report

The objective of the study was to evaluate the applicability of Lactobacilus sakei for the biopreservation of fresh gutted sea bass and sea bream packed under vaccum.

Is the term Lactilactobacillus sakei correct? Please revise carefully the microorganisms names and overall terminology throught the manuscript.

Is the final product a fermented fish? In this case it cannot be considered as fresh fish, but as a minimally proceed fish product.

How was low temperature achieved? The deviation 4±2°C is very high and should be justified.

It is weird that MAP (a well established methodology for fresh Mediterranean fish such as gilthead seabream and sea bass) did not provide inhibition of microbial growth, compared to VP, attributed to the bacteriostatic activity of CO2. I am afraid that 2 sampling points (additional to the initial point) are not adequate to provide appropriate results.

The experimental design should focus on the evaluation of the preservative effect of L.sakei including more sampling points and a more systematic shelf life study.

In general, the authors should elaborate more on the target preservation method (as also referred in the title of the manuscript) and justify the originality of the study.

Author Response

See the upload below

Reviewer 4 Report

The aim of the study should be more clearly described, especially in the abstract, which should contain more information about the results. Furthermore, the authors must rephrase/ rewrite many paragraphs and many scattered phrases throughout the manuscript. More importantly, results should be discussed and described in a more accurate way to make them clearer to the reader.

Author Response

See the upload below

Round 2

Reviewer 3 Report

The authors addressed adequately the reviewers' comments and the manuscript is significantly improved. However, a careful revision of the manuscript should be performed, mainly regarding the reported storage conditions (for example in the title the authors report storage at 6 degrees while in the manuscript 4 degrees C are indicated). To my opinion, the overall originality of the study remains questionable (e.g. effect of VP and MAP on the shelf life of sea bream and sea bass) and the authors should carefully justify the added value of the present study to the already published relevant studies.

The authors answers to the reviewers' suggestions should be clearly addressed not only to the response to the reviewers letter but most importantly throughout the revised manuscript.

Reviewer 4 Report

The authors have revised successfully the initial manuscript and responded sufficiently to all previous comments. Although, the originality of the paper is not substantial, some minor changes need to be taken into consideration .
